# Optimization of the Electrodeposition of Gold Nanoparticles for the Application of Highly Sensitive, Label-Free Biosensor

**DOI:** 10.3390/bios9020050

**Published:** 2019-03-31

**Authors:** Hao-Chun Chiang, Yanyan Wang, Qi Zhang, Kalle Levon

**Affiliations:** 1Department of Chemical and Biomolecular Engineering, NYU Tandon School of Engineering, Six Metrotech Center, Brooklyn, NY 11201, USA; hcc293@nyu.edu (H.-C.C.); qz260@nyu.edu (Q.Z.); 2State Key Laboratory of Precision Measuring Technology & Instruments, School of Precision Instruments and Optoelectronics Engineering, Tianjin University, Tianjin 300072, China; yanyanwang@tju.edu.cn

**Keywords:** electrodeposition, potentiometry, gold nanoparticle, nanoparticle size, protein binding, enzyme activity, optimization

## Abstract

A highly sensitive electrochemical biosensor with a signal amplification platform of electrodeposited gold nanoparticle (AuNP) has been developed and characterized. The sizes of the synthesized AuNP were found to be critical for the performance of biosensor in which the sizes were dependent on HAuCl_4_ and acid concentrations; as well as on scan cycles and scan rates in the gold electro-reduction step. Systematic investigations of the adsorption of proteins with different sizes from aqueous electrolyte solution onto the electrodeposited AuNP surface were performed with a potentiometric method and calibrated by design of experiment (DOE). The resulting amperometric glucose biosensors was demonstrated to have a low detection limit (>50 μM) and a wide linear range after optimization with AuNP electrodeposition.

## 1. Introduction

Label free biosensors with potentiometric signal generation and transduction are of great interest because they would not only significantly decrease the cost and time needed for sample preparation but also provide an easy-to-use tool to monitor binding events [1,2,3,4,5,6]. The principle of potentiometry is measuring the change of surface potential (working electrode) against to constant surface potential (reference electrode). Nowadays, potentiometric devices such as field effect transistor (FET) and light-addressable potentiometric sensor (LAPS) are the most trendy and sensitive electrical biosensors [7,8,9,10,11,12,13]. For the fabrication of these biosensors, the process of surface preparation is critical for stability, sensitivity, and safety in the application of clinical diagnostics [14].

For the determination of the high sensitivity of the biosensor, two major challenges are of key importance: (i) amplification platform and (ii) amplification process. So far, most of those reported researches mainly use the method of amplification process that utilizes nanoparticles as labels for signal amplification because of their unique electronic, catalytic and optical properties [15,16,17]. For using metal nanoparticles as the amplification platform, various approaches including self-assembly or grafting reaction have been achieved [18,19,20]. But a problem for these methods is that the preparation of films with more than 25% surface coverage is difficult due to the repulsive force between surface-confined nanoparticles and free nanoparticles in solution [21]. Sakai et al. purposed a method of electrochemical deposition which provides an easy and rapid alternative for preparation of nanoparticle platform in a short time with high AuNPs surface coverage [22]. Among the nanoparticles, gold nanoparticles (AuNPs) can be easily conjugated with biomolecules and retain the biochemical activity of tagged biomolecules, rendering AuNPs to be attractive materials for biorecognition applications. The detectable electron dense core of AuNPs with the high surface to volume ratio, and the controllable electrochemical behavior have all made AuNPs been widely used as sensitive tracers for biomolecular recognition events. [23,24,25]. Thus, in this work, the AuNPs were selected to form a thin film on the substrate of electrode with electrodeposited method, to act as an amplification platform for high sensitivity detection of biomolecules with potentiometric methods. As we control the size of the nanoparticle, we can monitor protein interaction with the particle of different size by potentiometer. The systematical study regarding preparation of AuNPs biofilm has great interests in the application of potentiometric sensor.

In electrodeposition, the size of nanoparticle has substantial effects on protein structure and stability compared to relatively “flat” supports [25,26]. For the flat substrates with nanoscale roughness, the effect of AuNPs size on protein interaction has not been widely studied. Here, we investigate three kinds of proteins of distinctly different sizes, but with similar isoelectric point (pI) values: Bovine serum albumin (BSA) is a triangular prismatic protein with a size of 14 nm × 4 nm × 4 nm (Mw 66.3 kDa, pI = 4.8); Glucose oxidase (GOx), a dimeric globular protein having overall dimensions of 6 × 5.2 × 7.7 nm (Mw 160 kDa, pI = 4.2) [27]; and casein proteins involved four types of proteins, alpha(s1)-casein (38%), alpha(s2)-casein (10%), casein (36%) and kappa-casein (13%), which form hydrated casein micelles about 100–300 nm in size (pI = 4.6) [28]. Layers of AuNPs with different sizes were fabricated to study the influence of the sizes of these three proteins to form the functional surface on the electrode. 

The purpose of this study is to understand how the parameters of electrodeposition for the preparation of the nanoparticle surface improve the performance of biosensor. AuNPs with different sizes were directly electrodeposited and compared onto glassy carbon (GC) and indium tin oxide (ITO) film coated glass slid. The surfaces of the modified electrodes were confirmed using Atomic Force Microscopy (AFM). The application of AuNPs modified electrodes was demonstrated for the detection of enzyme activity with electrochemical method.

## 2. Materials and Methods

### 2.1. Materials

Gold (III) chloride trihydrate 99.9%, BSA, GOx, and casein were purchased from Sigma-Aldrich and used without further purification. All solutions were prepared in Milli-Q water (18 ΩM). A pH 7.4 phosphate-buffered saline (PBS) solution of 0.2 M disodium orthophosphate (Na_2_HPO_4_), 0.2 M sodium dihydrogen orthophosphate (NaH_2_PO_4_) was prepared. Before used, the PBS solution was diluted to 10 mM with Milli-Q water. This PBS solution was used as electrolyte solution for potentiometric detection, and to prepare various protein solutions, which were stored at 4 °C while not in use.

### 2.2. Electrodeposition of AuNPs

The fabrication of AuNPs layer on GC electrode (3 mm dia., CH Instrument) was carried using electrochemical deposition [29,30,31]. GC electrode was polished with aluminum oxide powder and electrochemically polished in 1 M H_2_SO_4_ solution to remove any organic binders and contamination that occurs at electrode surface. ITO electrode was sonicated in acetone, ethanol, and distilled water (DI water) for 15 min consequently. After cleaning, the GC electrode was immersed into the solution of HAuCl_4_ in H_2_SO_4_. A cyclic voltammetric mode with the potential range of 1 V to −1 V was performed for electrodeposition. Then the electrode was quickly taken out, washed with DI water and dried with a stream of nitrogen. A multimode scanning probe microscope equipped with the type EV scanner and Nanoscope IIIa controller (Digital Instruments, Veeco) was used to image AuNPs as well as proteins post-treated on ITO surface in AFM.

### 2.3. Electrochemical Techniques

Potentiometric detection was performed by EMF interface instrument (Malvern, PA, USA) for monitoring potential change simultaneously in real time. The AuNPs modified GC working electrode and the Ag/AgCl (1 M KCl) reference electrode were immersed into the PBS solution and the signal of potential change was recorded. After the potential signal was stable (less than 1 mV drift in 10 min), a solution of proteins was added into the solution to check the signal changes for the characterization of biofilm [32]. The amperometric detection of glucose was performed in an electrochemical cell filled with 20 mM of 10 mM PBS at room temperature. In steady-state amperometric experiment, the potential was set at 0.6 V under gentle magnetic stirring [33].

### 2.4. Atomic Force Microscope Imaging

The AFM instrument was a multimode scanning probe microscope equipped with the type EV scanner, and Nanoscope IIIa controller (Veeco Instruments, Woodbury, NY, USA). Images were obtained in air at ambient temperature and humidity, within a range of 30–60% relative humidity. The tapping mode was employed. Image was analyzed by Veeco Instruments software for AuNP coverage calculation and using MATLAB image processing was performed for AuNP size quantification with a custom written code. The connected Component Analysis algorithm was applied in particle size measurement. The procedure of image processing is as follow: The image was converted to a binary image using Otsu’s thresholding. The closing process with disk structuring elements of 2 pixels was applied to remove noise included in the AFM image and enhance the edge. Each individual dot was then detected by using the Connected Component Analysis algorithm in the Image Processing Toolbox of MATLAB. The small unfilled objects of less than 4 pixels were subsequently removed. The filling process was required because the intensity of pixels surrounding the object was non-uniform. The sensitivity of Otsu’s thresholding was optimized by maximizing average particle size, since exceeding threshold would cause open circle and reduce the mean particle size, which is shown in Appendix A.

## 3. Results and Discussions

### 3.1. Electrodeposition Process

Figure 1 shows the CV plots of the electrodeposition of the AuNPs within different potential ranges. As seen, on the cathodic–going scan, the cathodic peak appeared at around ca. +0.4 V and is due to reduction of gold (III) to gold. This peak in the second cycle has shifted to more positive potential, which means easier electrodeposition of gold on the existing gold particles. With scanning to more negative potentials, a sharp increase in reduction current at potential more negative than −0.5 V is attributed to the reduction of water, resulting in the formation of hydrogen gas [34]. On the anodic–going scan, the peak at ca. +1 V is corresponding to the surface oxidation of the electrodeposited gold. Our AFM results also confirmed that the AuNPs was successfully deposited onto the surface of electrode with a quite symmetric distribution. From these results, we found that the electrodeposition of the AuNPs within different potential ranges exhibit different changing gold reduction peaks at different potentials. The peaks move towards more negative potential gradually along with extending of the potential range from 1 ~ 0 V to 1 ~ −0.4 V, 1 ~ −0.5 V and 1 ~ −1 V (Figure 1).

We then detected the protein adsorption ability of fabricated AuNPs modified GC electrodes by potentiometric method. The more negative potential applied during the electrodepositing, the more proteins are binding on the AuNP surface (Figure 1 inset). These results suggest that a low potential applied in the electrodeposition is essential for the controlled growth of the AuNP. Roustom et. al. reported that when preparing nanoscale-sized particles deposition onto electrode, nucleation and growth are the basic process should be involved [35]. If the low potential is above −0.4 V, the gold becomes hard to be nucleated. At potential more negative than −0.4 V, a nucleation with a sufficient large overpotential can be achieved to effectively seed the surface with nuclei [30]. So, in the following cycle, AuNPs can start to grow on the surface of electrode. When the low potential is more negative, the processed of nucleation and growth may take place simultaneously. In this study, we applied −1 V potential to increase the growth rate of AuNPs.

### 3.2. The Modification of AuNPs Electrode

It is well known that there are three mechanisms to explain the adsorption of protein to AuNPs: electrostatic interaction of AuNPs and opposite charged proteins; covalent bonding between the thiols/amine group present within the amino acids in the protein and AuNPs; and hydrophobic interaction between proteins and AuNPs. Global electrostatic effects may dominate when the protein is structurally stable and the solid surface is hydrophilic [36]. As BSA, GOx and casein are negatively charged in the neutral PBS buffer (pH 7.4), and the difference of electrostatic interactions between proteins and AuNPs can be neglected. Figure 2 shows the potentiometric response of these three kinds of AuNPs modified GC electrodes. When the AuNPs modified electrodes contact with proteins solutions, negative shift of the potential were instantly observed and followed a pattern of stepwise increase upon further titrations. The potential responses were caused by the contribution of adsorbed proteins. To verify the signal amplification by AuNPs, we compared the potentiometric response of these three kinds of proteins adsorption on GC and AuNPs modified GC electrodes. From Figure 2, we can see that there was almost no potential change with non-modified GC electrode (curves A, B and C). Assumedly small proteins adsorption on GC electrode occurs, or because of the much worse conductivity and smaller surface area of GC, it hardly sensed any potential response when protein adsorption occurred. This proves that the AuNPs form an excellent platform for increased proteins loading, and efficiently improving the charge transfer between analyte and the electrode surface. The results also indicated that different kinds of proteins have different bonding behavior with AuNPs, which is size dependent, and the potential change is corresponded to the amount of protein adsorption [9,33].

### 3.3. Deposition of Proteins on AuNP Covered Electrode Surface

For the mechanism of the electrodeposition of the AuNPs, it was confirmed that free gold (III) ions from solution will be attached to the surface of the electrode via electrostatic interaction first, then the application of potential to electrode promoted the subsequent reduction of gold (III) ion [37]. The size and quantity of AuNPs electrodeposited on the electrode surface relies on the gold (III) ion absorbance and deposition time. The concentration of HAuCl_4_, acidity of the solution media, scan cycles and scan rate also have an effect on the size and film thickness of AuNPs deposited. Here, we electrodeposited AuNPs films on electrodes by cyclic voltammetry, and controlled the growth of nanoparticles size and film thickness with these different parameters. Three kinds of proteins, casein, GOx and BSA, which have different molecular weight with the similar value of pI were used for the evaluation of the adsorption of the proteins. As all the pIs of BSA, GOx and casein are around 4.5, these proteins have similar negative charges in PBS buffer (pH 7.4). In this part, the change of potentiometric was measured to value their ability of adsorption on the surface of AuNPs. Our goal was to understand how the formation of AuNPs layer affects the adsorption of the proteins. 

Figure 3 shows the potentiometric difference due to the electrodeposition parameters of (a) HAuCl_4_ concentration, (b) H_2_SO_4_ concentration, (c) scan cycles, and (d) scan rate on the response of AuNPs modified GC electrode toward to 10 μg mL^−1^ of casein, GOx and BSA adsorption respectively, 16 cases were performance for each protein. According to the guidance of AuNPs layer preparation using electrochemical method, the growing conditions are linear to the size of deposited nanoparticle within the applied range [38]. Interestingly, in Figure 3 we observed the potentiometric difference is not a direct proportional relationship to all the parameters, which refers the size is not the main factor for optimal potentiometric. Design of Experiment (DOE) analysis was applied to find the optimal region of electrodeposition for potentiometric measurement, the description of DOE work is shown in Appendix A. The perspective region of potentiometric difference of BSA for the adsorption on AuNPs was observed when AuNPs was electrodeposited in the solution containing 0.5 M HAuCl_4_ (Figure 3a▲) and 0.5 M H_2_SO_4_ (Figure 3b▲) with 5 cycles (Figure 3c▲) at the scan rate of 50 mV s^−1^ (Figure 3d▲) (called method 1). In the following, we detected the GOx and casein adsorption efficiency with different sizes and densities of AuNPs. We found that the optimization of GOx bonding on AuNPs was observed when AuNPs was electrodeposited in the solution containing 1mM HAuCl_4_ and 0.5 M H_2_SO_4_ with 2 cycles at the scan rate of 20 mV s^−1^ (called method 2). While for casein bonding, the condition for the maximum bonding is 2 mM HAuCl_4_ and 1 M H_2_SO_4_ with 15 cycles at the scan rate of 50 mV s^−1^ (called method 3). The results indicate that different sizes of proteins have different adsorption abilities to AuNPs as well as suggests optimal conditions for maximizing responses due to formation of protein biofilm. For example, 1mM HAuCl_4_ solution can generate AuNPs layer to form maximum GOx coverage according to Figure 3a. The particle size of AuNPs controls the nanoscale roughness of AuNPs layer on GC electrode in which protein can be adsorbed in a good condition to form a continuous layer without any additional passive layer, such as 6-Mercapto-1-hexanol (MCH) [39]. The reduction of potentiometric signal in the condition of formation of larger AuNPs may be caused by the association change between the AuNPs layer and proteins due to the increased size of AuNPs. The following AFM results provide more information of surface morphology and the model of protein adsorption.

### 3.4. AFM Results and a Comparative Model

The morphologies of the modified electrodes were investigated by AFM. For the convenience of AFM operation, AuNPs were electrodeposited on the ITO electrodes with the methods 1, 2 and 3 (described in the previous paragraph). Figure 4 presents the AFM scans obtained over 2 μm × 2 μm areas and plotted using the same scale in order to facilitate comparisons. The vertical scale ranges are from 0 to 125 nm (deep red to white). The area of ITO surface was determined using Bearing analysis and the coverage of AuNPs can be calculated (Appendix A) Figure 4a–c show the coverage of electrodeposited AuNPs of 99%, 99%, and 97% using the methods 1, 2 and 3. We can see from Figure 4a–c, that films of AuNPs were formed on the ITO surface. The Connected Component algorithm was used to obtain the mean sizes of the AuNPs [40], which were shown to be 52, 91 and 184 nm for the surfaces fabricated using methods 1, 2 and 3, respectively. The step of analysis is also described in Appendix A. These results confirm that the sizes of synthesized AuNPs depended on solution condition, cycles and scan rate. Thus, different AuNPs sizes could be made by the method of electrodeposition with different parameters. More important is that the result also suggested the binding of proteins were associated with the sizes of AuNPs. The sizes of proteins are in the order of Casein > GOx > BSA. It means that the smaller the proteins are, the easier adsorption on smaller sizes of AuNPs is (Scheme 1). This result also has a good agreement with Lacerda et al.’s AuNPs-protein interaction report, which indicated that the binding association constant of Albumin is optimal in AuNP size of 60 nm and best association for Globulins (~155 kDa, similar to GOx’s 160 kDa) happens when AuNP size is over 80 nm [41]. After proteins adsorbed on those AuNP modified surfaces, higher features and bigger particle sizes were also observed (Figure 4d–f). In Figure 5, the optimal potentiometric response for each protein is plotted as a function of AuNP size (▲), which suggests that the size of protein would affect the sensitivity of potentiometric measurement. In the same plot, the coverage of AuNP (■) shows indifference to the potentiometric measurement. In the following, GOx was selected to fabricate a biosensor for verify the enzyme activity and determine the sensitivity of biosensor application. 

### 3.5. Glucose Characterization on GOx-Modified AuNP-Surface 

Proteins are highly surface active and they interact with solid-liquid interfaces mainly through three subprocessed: structural rearrangement in the protein molecule [41,42]; dehydration of parts of protein and surface hydrophobic effect; and redistribution charged group in the interfacial layer [36]. If proteins are bound to the solid surface, most of proteins will undergo denaturation of their tertiary structure and their secondary structure could also be disrupted in some case. We evaluated the activity of GOx after them adsorbed on the surfaces of AuNPs [33]. 

GOx can catalyze the oxidation of β-D-glucose to D-glucono-δ-lactone and hydrogen peroxide which then can be detected by an amperometric method. Figure 6 illustrated a typical steady—state response of GOx modified GC electrode on successive addition of different concentration of glucose. It presents a linear response to glucose concentration within the range from 0.05 mM to 8 mM and a high sensitivity of 2.08 × 10^−3^ A·M^−1^·cm^−2^ (R = 0.98). The remarkable current increase confirmed that GOx kept its activity on the AuNPs surface. These results indicated that GOx largely retains its native-like structure. The high sensitivity of this detection can be attributed to the AuNPs, which provide a biocompatible microenvironment to maintain the activity of the protein and greatly enhance the conductivity.

## 4. Conclusions

The preparation of a label-free highly-sensitive biosensor using the electrodeposition of AuNPs was systematically studied and optimized. The sizes of AuNPs were carefully controlled by electrolyte conditions; HAuCl_4_ and acid concentrations, and scan cycles and scan rates, during the electrodeposition. AFM imaging was used to confirm the particle sizes and GC surface treatment, which allowed us to propose a deposition model for protein binding. The adsorption of three model proteins with different sizes, BSA, GOx and casein, on the surfaces of AuNP modified electrodes were investigated. It is found that the sizes of the AuNPs played an important role on the adsorption of the proteins; proteins with different sizes have different surface coverage on AuNPs, being a size dependent phenomenon. The optimal conditions for the electrodeposition of the AuNPs were able to maximize the protein binding on the coverage of AuNPs without extra treatment, such as MCH, as a passive layer. 

An application of GOx modified AuNPs electrodes was demonstrated as optimized biosensing with the ability to quantitatively detect glucose with alow detection limit (>50 μM) and a wide linear range. The result presented here are not only significant to basic science research but also important to understand potential application of these devices as label-free sensing platform.

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
