# Peer review of "Optimization of the Electrodeposition of Gold Nanoparticles for the Application of Highly Sensitive, Label-Free Biosensor"

_biosensors, 2019, doi:10.3390/bios9020050_

Round 1

Reviewer 1 Report

In the work „Optimization of electrodesposition of gold nanoparticles for the application of highly sensitive, label-free biosensors” interesting results of studies were presented. Generally, the authors focused on the preparation of gold nanoparticle layers on the surfaces of electrodes using electrodeposition process. Then, the processes of adsorption of three types of proteins on these gold surfaces were carefully investigated applying electrochemical technique.  In the improved version of manuscript, especially in the Supporting materials, one can find insightful analysis of properties of electrodeposited gold nanoparticle layers as well as description of optimization of processes of formation of such layers.

In my opinion, this supplemented version of work is suitable for publication in the journal “Biosensors” and I believe that this work will be interesting and useful for a broad scientific community.

Author Response

To Reviewer 1 Thank you for your time and effort to go through our paper and submit your review

To Review 2  Thank you for this useful suggestion for the experiment section in the manuscript, we added the detail of measurement of AuNP size in line 104 – 115:

using MATLAB image processing for AuNP size with a custom written code. Connected Component Analysis algorithm is applied in particle size measurement. The procedure of image processing as follow: The image was converted to a binary image using Otsu’s thresholding. The closing process with disk structuring elements of 2 pixels was applied to remove noise included in the AFM image and enhance the edge. Each individual dot was then detected by using the Connected Component Analysis algorithm in the Image Processing Toolbox of MATLAB. The small unfilled objects of less than 4 pixels were subsequently removed. The filling process was required because the intensity of pixels surrounding the object was non-uniform. The sensitivity of Otsu’s thresholding was optimized by maximizing average particle size, since exceeding threshold would cause open circle and reduce the mean particle size, which is shown in Figure S5 of supporting information.

Reviewer 2 Report

Dear authors, 

I carefully evaluated the manuscript (biosensors-460726) entitled “Optimization of Electrodeposition of Gold Nanoparticles for the Application of Highly Sensitive, Label-Free Biosensor” submitted for publication in “Biosensors”.

In this paper the authors presented a highly sensitive biosensor nanostructured with gold nanoparticles (AuNP). As authors clearly demonstrated, the size of the AuNP was critical for the performance of the sensor since it affected the adsorption of proteins of different sizes on the sensor surface. This is the core of the paper. 

The analytical approach was excellent and the paper is written in good English. The title clearly describes the article. The abstract reflects the content and the goal of the manuscript. 

The Introduction is exhaustive, the objective is clear as well as the approach to reach it, and the reported literature was sufficient, helpful and definitely relevant to the purpose of the manuscript. Results are clearly reported and, in accordance with the aims of the manuscript, each point is well discussed and compared with the previous literature. Figures and captions are clear, exhaustive and detailed. 

Generally speaking, the manuscript is good but, it is reviewer’s opinion, that the experimental section has a weakness. In the abstract, in the introduction and in the results and discussion sections, the role of size of AuNPs was emphasized but, in the experimental section, the only reference to AuNP size is at lines 79-80 and every detail about AuNPs was transferred to “Supporting Information”. This latter section was properly done but more detailed information about AuNPs size and AuNps coverage should be made explicit in the Experimental section.

According to the previous consideration, it is reviewer’s opinion that the manuscript needs a revision of the experimental section to be accepted for publication on Biosensors.

Author Response

To Review 2  Thank you for this useful suggestion for the experiment section in the manuscript, we added the detail of measurement of AuNP size in line 104 – 115:

using MATLAB image processing for AuNP size with a custom written code. Connected Component Analysis algorithm is applied in particle size measurement. The procedure of image processing as follow: The image was converted to a binary image using Otsu’s thresholding. The closing process with disk structuring elements of 2 pixels was applied to remove noise included in the AFM image and enhance the edge. Each individual dot was then detected by using the Connected Component Analysis algorithm in the Image Processing Toolbox of MATLAB. The small unfilled objects of less than 4 pixels were subsequently removed. The filling process was required because the intensity of pixels surrounding the object was non-uniform. The sensitivity of Otsu’s thresholding was optimized by maximizing average particle size, since exceeding threshold would cause open circle and reduce the mean particle size, which is shown in Figure S5 of supporting information.

Round 2

Reviewer 2 Report

In the revised manuscript the authors addressed properly the topic proposed by the reviewer and, with respect to the previous version, the quality of the manuscript improved. I suggest to accept the manuscript.

This manuscript is a resubmission of an earlier submission. The following is a list of the peer review reports and author responses from that submission.

Round 1

Reviewer 1 Report

The authors have done a great job optimizing the electrode position of gold nanoparticles on a GC electrode. The manuscript however suffers from several limitations before it could be published. 

First the article should be edited as there are quite some grammatical errors that need to be fixed. For instance line 44, the phrase should be rewritten as its not grammatically correct. Similarly, line 116-122 Contain several mistakes.

The main important comment however is, there are many studies on CV electrodeposition of gold nanoparticles for immunoassay purposes, what is the benefit and advantage of this study over them. I can't see any protocol or guideline suggesting which particle size should be addressed based on the size of the target protein. 

Line 112-122 the authors have discussed some results but it would be more interesting if they also mention the size of these particles

Later on line 146, they mention the bonding behavior is size dependent. This is while until here no mention of the size of the particles could be found. So more explanation should be provided for this clause.

Line 167-184: the authors have mentioned that the maximumof 10 mcg/ml of BSA was adsorbed on the particles. I cant find any results showing how they measured or confirmed that all the antibody was adsorbed. Aggain here i am missing the size of the particles.

Line 172 when you are discussing each protein separately, i dont know whats the reason of pointing it out only for GOx and casein, the same is true for BSA.

Line 182 how do you know that a continuous layer with no passive layer is formed. 

The authirs have studied the efffect of HAuCl4 and H2SO4 on the particle size separately. This is while their ratio is alsoof great importance.

Does the potential range in which the electrodeeposition happens affect the particle size?

line 220-227 you have only discussed GOx and concluded that BSA also retains its functionality after immobilisation. The results are missing. In the same section, you have reported the results for a specific particle size. It would have been more interesting if the same results for other particle sizes would have been presented and then you could show that this size provided the best results.

The results for casein is also missing

References. Certain references contain multiple references. They should be separated. For instance reference 4, 9

Author Response

First the article should be edited as there are quite some grammatical errors that need to be fixed.

We have edited and corrected with our best effort, all answers are in bold italics.

For instance line 44, the phrase should be rewritten as its not grammatically correct. 

The detectable electron dense core of AuNPs with the high surface to volume ratio, and the controllable electrochemical behavior have all made AuNPs been widely used as sensitive tracers for biomolecular recognition events.

Similarly, line 116-122 Contain several mistakes.

The long sentence has been modified to two separate sentences, both with more clarity.

The main important comment however is, there are many studies on CV electrodeposition of gold nanoparticles for immunoassay purposes, what is the benefit and advantage of this study over them.

The advantage is that with the change in the deposition, we can control the size of the particle. Then, with the control of the size, we can also use potentiometry to monitor the binding of a protein on the AuNP surface.

We added the sentence:

As we control the size of the nanoparticle, we can monitor protein interaction with the particle of different size by potentiometer.

I can't see any protocol or guideline suggesting which particle size should be addressed based on the size of the target protein. 

The goal for this is addressed on lines 62-63:

Films of AuNPs with different sizes were fabricated to study the influence of the sizes of these three proteins to form the functional surface on the electrode.

Line 112-122 the authors have discussed some results but it would be more interesting if they also mention the size of these particles

The sizes of the particles will be discussed later with the AFM characterization. Now we added in parenthesis to the abstract: (we selected sizes of 5, 14 and 40 nm) .

Later on line 146, they mention the bonding behavior is size dependent.

This is while until here no mention of the size of the particles could be found. So more explanation should be provided for this clause.

Again, the sizes were measured with AFM with a shown size distribution. We selected the sizes of 5, 14 and 40nm for the further experiments

Line 167-184: the authors have mentioned that the maximum of 10 mcg/ml of BSA was adsorbed on the particles. I cant find any results showing how they measured or confirmed that all the antibody was adsorbed. Again here i am missing the size of the particles.

We selected the sizes of 5, 14 and 40nm for the further experiments

Figure 3 shows our design to standardize the size ratio based on potentiometric measurements

Line 172 when you are discussing each protein separately, i dont know whats the reason of pointing it out only for GOx and casein, the same is true for BSA.

Now All BSA, GOx and Casein are mentioned together

Line 182 how do you know that a continuous layer with no passive layer is formed. 

The binding only occurs with proteins on gold.

The authirs have studied the efffect of HAuCl4 and H2SO4 on the particle size separately. This is while their ratio is alsoof great importance.

We measured each effect independently keeping others constant and optimized from potentiometric responses. Investigation of the the ratio effect would have been much more complex.

Does the potential range in which the electrodeeposition happens affect the particle size?

The potential is a function of the protein binding on the particle, the potential does not effect the sizes.

line 220-227 you have only discussed GOx and concluded that BSA also retains its functionality after immobilisation. The results are missing.

Mentioning BSA removed form this enzyme analysis section

In the same section, you have reported the results for a specific particle size. It would have been more interesting if the same results for other particle sizes would have been presented and then you could show that this size provided the best results.

The segment focuses on enzyme activity with the selected sizes of 5, 14 and 40nm, which were already selected as the most optimal sizes. Yes, repeating the size dependence with enzyme reaction would be an effective control

The results for casein is also missing

Fig 1-4 present results for all three, (casein not in the enzyme section)

References. Certain references contain multiple references. They should be separated. For instance reference 4, 9

Ref 4 contains surface studies of Author’s group

Ref 9 contains surface measurements by Author’s group

Reviewer 2 Report

Optimization of Electrodeposition of Gold Nanoparticles for the Application of Highly Sensitive, Label-Free Biosensor

Yanyan Wang, Hao-Chun Chiang, Qi Zhan and Kalle Levon

The manuscript describes the electrodeposition method for fabricating gold nanoparticle electrodes for biosensing. Various biomolecules were investigated in this study. The authors also observed that optimal interaction of the biomolecules with the underlying gold nanoparticles is dependent on the deposition conditions. The manuscript is well written, however there are some concerns and questions that need to be addressed. If these questions and suggestions are addressed, then the manuscript is fit for publication.

Line 24: please check sentence construction; not only....but also

Line 78: Please cite recent references on electrodeposition of Au. Please use the following reference. This addresses the size and density of Au NPs with deposition conditions.

Journal of nanoscience and nanotechnology 18 (5), 3492-3498

line 79: Please briefly explain, what is electrochemically treated?

Line 113: Please correct sentence construction. “It seems” is less scientific.

Line 121: Please check sentence construction.

Fig 1: The Y axis can be changed from A to mA, to maintain consistency with the inset.

Line 134: There is a typo: figure 2

Fig 2: Please indicate (with an arrow) when protein was added during the experiments. Was protein added at certain time intervals?

Author Response

Answers/corrections in bold italics

Line 24: please check sentence construction; not only....but also

 but also provide an easy-to-use tool to monitor binding events”

Line 78: Please cite recent references on electrodeposition of Au. Please use the following reference. This addresses the size and density of Au NPs with deposition conditions.

Journal of nanoscience and nanotechnology 18 (5), 3492-3498

We have added new references for the AuNP size and density studies.

line 79: Please briefly explain, what is electrochemically treated?

Changed to …polished…

Line 113: Please correct sentence construction. “It seems” is less scientific. = line 120

The more negative potential applied during the electrodepositing, the more proteins are binding on the AuNP surface

Line 121: Please check sentence construction.

These results suggest that a low potential applied in the electrodeposition is essential for the controlled growth of the AuNP.

Fig 1: The Y axis can be changed from A to mA, to maintain consistency with the inset.

Changed accordingly

Line 134: There is a typo: figure 2

Extra “modified” removed

Fig 2: Please indicate (with an arrow) when protein was added during the experiments. Was protein added at certain time intervals?

Figure caption states: 1, 10, and 100μg mL-1 of BSA, GOx and casein were added in series

Reviewer 3 Report

The work entitled „optimization of electrodeposition of gold nanoparticles for the application of highly sensitive, lable-free biosensors” presents interesting results of studies. Nevertheless, at the moment I cannot support it publication in the Biosensors.

First of all, reading the manuscript I have a feeling that the work presents only qualitative research results whereas a little attention is paid on the quantitative description of the processes which were observed. Numerous parameters which can influence on the adsorption of the proteins on the modified electrodes were not take into account. In the introduction the authors mentioned that the coverage of gold nanoparticle monolayers formed in self-assembly processes does not exceed 25%. As they noticed it is an important parameter which influence on the formation of the top layer (bilayer) e.g. proteins. However, the coverage of gold nanoparticle layers was not determined in the quantitative manner. Indeed, the authors showed the AFM images of the formed gold nanoparticle layers therefore one can see their structure and homogeneity but the coverage is unknown. One can see that it is really high and I am afraid that it is too high to determine it in a quantitative manner basing on the AFM images (in the case of such high coverage the tip does not measure (divide out he particles). Therefore, I have to advice imaging with the use of scanning electron microscopy (SEM) in order to determine the coverage of layers. Secondly, from the high profiles the authors should determine the thickness of the deposited films. Is it possible that the electrodeposition caused formation of bi- or multilayers? In turn, analyzing the AFM images recorded for bare gold nanoparticle layers and for the layers with deposited proteins it is impossible to see the differences. Maybe the authors should present high profiles.

Misleading interpretation of the experimental results is related to the influence of nanoparticle size on the adsorption process of proteins. It is well known that roughness of such films plays a pivotal role in the adsorption process of proteins. I advice to get acquainted with the work “Protein adsorption mechanisms at rough surfaces: Serum albumin at a gold substrate”.

Considering the available literature reports devoted to the deposition of proteins on nanoparticle monolayers I am not convince that the results of studies presented in this manuscript bring a novelty in the stage of knowledge referring to the production of biosensors from gold nanoparticles. In my opinion achieved results are not groundbreaking in order to publish them in the journal Biosensors.

Author Response

Our answers and corrections in bold italics

First of all, reading the manuscript I have a feeling that the work presents only qualitative research results whereas a little attention is paid on the quantitative description of the processes which were observed.

The application of our tools is to show quantitative use of reduction of ionic Au salts to solid Au nanoparticles with quantitatively controlled size. As a follow-up we show the use of potentiometry for the quantitative monitoring of protein binding on gold.

Numerous parameters which can influence on the adsorption of the proteins on the modified electrodes were not take into account.

We show quantitatively the adsorption of protein on different sizes of AuNP while many factors (affinity, curvature etc) may affect the result.

In the introduction the authors mentioned that the coverage of gold nanoparticle monolayers formed in self-assembly processes does not exceed 25%.

That’s an estimate from the AFM measurements

As they noticed it is an important parameter which influence on the formation of the top layer (bilayer) e.g. proteins.

Yes, protein binding on gold particles

However, the coverage of gold nanoparticle layers was not determined in the quantitative manner.

But we show quantitatively the weak binding on the non-gold surface, thus our work focuses on protein-Au adsorption only

Indeed, the authors showed the AFM images of the formed gold nanoparticle layers therefore one can see their structure and homogeneity but the coverage is unknown.  

We focus on the protein binding on the AuNP sizes of 5, 14 and 40 nm

One can see that it is really high and I am afraid that it is too high to determine it in a quantitative manner basing on the AFM images (in the case of such high coverage the tip does not measure (divide out he particles).

Potentiometry only measures the protein binding on the AuNP

Therefore, I have to advice imaging with the use of scanning electron microscopy (SEM) in order to determine the coverage of layers. Secondly, from the high profiles the authors should determine the thickness of the deposited films.

The AFM pictures show clearly that the AuNPs do not cover the area totally. The 25% is only a visual estimate but on the other hand, we show that the protein binding on glassy carbon doesn’t give a potential change (fig2), thus we focus on protein binding on the AuNPs. It is a justified assumption as the results (Fig 3) show a very clear trend with our assumption.

In turn, analyzing the AFM images recorded for bare gold nanoparticle layers and for the layers with deposited proteins it is impossible to see the differences. Maybe the authors should present high profiles.

Figure 3 shows the optimization process, all different results with the different proteins = optimal for each protein

Misleading interpretation of the experimental results is related to the influence of nanoparticle size on the adsorption process of proteins.

As we use potentiometry to monitor the binding process, we show the dependence on the AuNP.

It is well known that roughness of such films plays a pivotal role in the adsorption process of proteins.

We selected the size as the variable and with potentiometric monitoring we show a clear dependence

I advice to get acquainted with the work “Protein adsorption mechanisms at rough surfaces: Serum albumin at a gold substrate”.

 Thank you, we have studies now the suggested article and still emphasize that our selection of the variable shows a quantitative dependence.

Considering the available literature reports devoted to the deposition of proteins on nanoparticle monolayers I am not convince that the results of studies presented in this manuscript bring a novelty in the stage of knowledge referring to the production of biosensors from gold nanoparticles.

In my opinion achieved results are not groundbreaking in order to publish them in the journal Biosensors.

The use of potentiometry to show the dependence of the AuNP size dependence of protein binding has not been shown as described in our effort.

Round 2

Reviewer 1 Report

The authors have tried really hard to address the comments addressed by the reviewers; however, I still think the presentation of the results is not clear.

The authors claim that they have been testing three different particle sizer but it’s hard to follow first of all why they have selected each size for the test and then really see the effect of the size in the results 

I would have expected them to conclude which particle size to use with which application or protocol 

Reviewer 3 Report

I support my initial assessment of the manuscript. The revised version of the manuscript does not include the improvements suggested by me. The answers are inadequate to my questions.